# Activation of Peroxymonosulfate by UV-254 nm Radiation for the Degradation of Crystal Violet

Noreen Ali [1], Ashfaq Ahmad Khan [1], Muhammad Wakeel [2,*], Ijaz Ahmed Khan [1], Salah Ud Din [3], Saeed Ahmad Qaisrani [2], Abdul Majid Khan [1] and Muhammad Usman Hameed [4]

1   Department of Chemistry, Women University of Azad Jammu & Kashmir, Bagh 12500, Pakistan
2   Department of Environmental Sciences, COMSATS University Islamabad, Vehari Campus, Vehari 61100, Pakistan
3   Department of Chemistry, University of Azad Jammu & Kashmir, Muzaffarabad 13100, Pakistan
4   Department of Chemistry, University of Poonch, Rawalakot 12350, Pakistan
*   Correspondence: wakeelrana@cuivehari.edu.pk

**Abstract:** Water is a precious natural resource. Unfortunately, bodies of water become polluted by waste, such as untreated wastewater and detritus, along with oil spills, with minimum or no consideration for their limited capacity to renew themselves. Among these pollutants, dyes are harmful as they are persistent and not biodegradable in nature. The present study demonstrates the removal of crystal violet (CV), a toxic cationic dye, by using three systems: Peroxymonosulfate (PMS), UV-254 nm radiation and UV/P5MS. The effects of various parameters, such as the effects of the initial dose of crystal violet, initial concentration of PMS, pH, typical inorganic ions, etc., were also investigated. The effect of pH was investigated in the range of 1.92–12.07. Similarly, the effect of various anions such as $NO_2^{\bullet-}$, $HCO_3^{\bullet-}$, $CO_3^{\bullet 2-}$, $SO_4^{\bullet 2-}$ and $CH_3COO^{\bullet-}$ was investigated for the degradation of target pollutants. The order of degradation of crystal violet was UV/PMS > PMS > UV with removal efficiencies of 97%, 76% and 42%, respectively, at reaction times of 60 min. The degradation of crystal violet was enhanced significantly at a pH range of 10.52–12.07. Electrical energy per order (EE/O) values for UV/PMS, PMS and UV were calculated to be 1.68, 3.62 and 48.96 KWh/m$^3$/order, respectively. The addition of inorganic ions inhibited the removal of CV in the order of $SO_4^{\bullet 2-}$ > $NO_2^{\bullet-}$ > $HCO_3^{\bullet-}$ > $CO_3^{\bullet 2-}$ > $CH_3COO^{\bullet-}$. Moreover, the kinetic studies on the degradation of CV by the UV-254 nm, PMS, and UV/PMS systems, were also carried out and found to follow pseudo-first-order kinetics. The study revealed that oxidation processes are most efficacious for the removal of organic dyes from wastewater.

**Keywords:** advanced oxidation processes (AOPs); crystal violet; degradation pathways; dyes; wastewater treatment

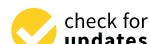



## 1. Introduction

The textile industry is a major source of dyes and about 15% of the dyes are wasted during synthetic processes and the treatment of industrial effluents, which pose a serious threat to marine life and are also harmful to human beings [1–3]. The residues of textile industries have aroused much concern due to their harmful effect on aquatic life as well as human beings and is usually considered one of the major coloring contaminants in water bodies [4]. The release of these textile effluents into water bodies has been reported to be carcinogenic and hazardous as they cause major health issues [5–7]. Among the broad range of triphenyl methane dyes, crystal violet is considered one of the most carcinogenic and toxic in nature [8] due to its complex aromatic structure [9,10]. This cationic dye, due to its mutagenic and non-biodegradable nature, is considered a major pollutant when directly discharged into water and poses a major threat to aquatic life as well as human life. The major health issues caused by CV are vomiting, diarrhea, headache, dizziness, as well as damage to the gastrointestinal tract by excessive ingestion of CV [11].

Crystal violet (methyl violet) is a cationic dye with basic monocovalent in nature, considered a mixture of three basic compounds in its structure commonly referred to as tetramethyl, pentamethyl and hexamethyl, which differ from each other on the basis of the number of methyl groups in their amine functional group [12]. There are various applications of CV in different fields like textile industries and pharmaceutical industries along with their diverse range of implementation in each industry, such as dye paper, printing inks, fertilizers, disinfectants, anti-freeze and leather jackets [13–15] for an intravenous agent, veterinary products, in the staining of biological process, a supplement of poultry feed to minimize the dispersion of fungus, forge and parasites of the intestine, industrial dying, the coloring of wood, silk, paper, and inks as biological stain paper [16], and in the cosmetic and food industries. In addition, triphenylmethane dyes are employed as staining agents in bacteriological and histopathological applications and for the coloration of textile products [17]. The removal of crystal violet from wastewater due to its toxicity and high solubility in water is thus necessary [18].

There are various classical methods used for wastewater treatments, including flocculation, reverse osmosis, biological methods, and adsorption and chemical methods, but there are some limitations related to these methods [19,20], such as flocculation, reverse osmosis and adsorption, and often transmit the organic contaminants to other media, thus leading to secondary pollution [21]. Similarly, chemical methods are usually high-priced and also the formation of sludge creates a large issue of removal. In addition, there is an unnecessary use of chemicals during this process which cause environmental pollution. In addition, large amounts of electricity and other chemicals are consumed during the process and are usually costly [22].

With the passage of time, various technological developments have been applied for the removal of these contaminants. Among these technologies, advanced oxidation processes (AOPs) are one of the most important processes, which are highly selective technologies for removing and demounting the means of disintegration of those pollutants, which are relatively stable in nature [22,23]. Currently, the advanced oxidation technologies (AOTs) used are environmentally compatible and capable of oxidizing a number of contaminants that rely on the production and use of hydroxyl radical (OH) [24] and sulfate radical ($SO_4^{\bullet -}$), which can more efficiently and selectively oxidize rapidly into $CO_2$, $H_2O$ and inorganic species by the process of mineralization to process an elevated number of organic pollutants [25,26].

One of the most important and highly efficient oxidants is peroxymonosulfate (PMS, $HSO_5^{\bullet -}$), which is considered an active agent of oxone in aqueous solutions with a general formula of $2KHSO_5 \bullet KHSO_4 \bullet K_2SO_4$. This formula shows that it has the triple salt of potassium, is also eco-friendly in nature, and is considered versatile in nature with high efficiency [27,28], which is one of the most important resources of highly reactive radicals, such as hydroxyl and sulfate radicals, respectively. The hydroxyl radicals ($^{\bullet}OH$) are the most powerful and non-selective oxidants and have a value of bimolecular rate constants of $10^7$ to $10^{10}$ $M^{-1}$ $s^{-1}$ for organic pollutants [29], a high standard redox potential such as ($E^{\circ}$ = 2.7 V) [30], and thus, carried out complete the degradation of organic chemicals and pollutants, which are difficult to detoxify with traditional approaches [31]. The sulfate radicals ($SO_4^{\bullet -}$) are the most active radicals with a standard redox potential of 2.5–3.1 V [32,33] and bimolecular rate constants for organic pollutants in the range of $10^5$–$10^9$ $M^{-1}$ $s^{-1}$ [34]. PMS can be activated by UV radiation into radicals, i.e., $^{\bullet}OH$ and $SO_4^{\bullet -}$, through the hemolytic breakage of the peroxide bond (-O-O-) [35,36].

The UV/PMS AOT is, relatively, the most successful and highly energetic technology for the decolourization and decontamination of aquatic environments due to their excellent water solubility, low cost, ease of storage, ease of transportation, low environmental impact, high rate of oxidation, simplicity, and no resultant sludge formation [36]. The activation of PMS by ultraviolet radiation is considered one of the most feasible methods [37] because of its activation energy [38,39]. The activation of PMS generates two types of radicals, such as $^{\bullet}OH$ and $SO_4^{\bullet -}$, which are the most efficient radicals for the degradation of

organic contaminants. As sulfate radicals, ($SO_4^{\bullet-}$) radicals are the most effective for the degradation of organic contaminants in aquatic environments with a high amount of natural organic matter (NOM) due to their relatively low tendency for NOM [36].

The degradation of crystal violet by ultraviolet radiation (UV-254 nm) is also highly effective due to the formation of hydroxyl radicals and hydrogen radicals due to the decomposition of water molecules [40,41]. Such radicals further degrade the organic contaminants into less toxic products. Owing to such factors, sulfate radicals (SR-AOTs) are considered the most effective for the degradation of carcinogenic and toxic organic contaminants in aquatic environments, such as dyes [42], pesticides [43], sunscreen agents and pharmaceuticals [44,45] into less toxic by-products.

This work is mostly focused on introducing the most successful, highly energetic and environmentally friendly treatments, such as hydroxyl and sulfate radicals, which are advanced oxidation technologies with different initial doses of oxidants as well as target organic contaminants. The effects of various parameters were also investigated for these systems, such as the effect of an initial dose of CV, initial concentration of PMS, pH, typical inorganic ions, etc. The effect of pH was investigated in the range of 1.92 to 12.07. Similarly, the effects of various anions such as $NO_2^{\bullet-}$, $HCO_3^{\bullet-}$, $CO_3^{\bullet2-}$, $SO_4^{\bullet2-}$ and $CH_3COO^{\bullet-}$ were investigated for the degradation of target pollutants. Total carbon (TC) analysis was carried out by ECOSAR. Degradation of by-products of CV were detected by HPLC-MS to establish new potential degradation pathways of CV. Kinetic study was carried out in the UV-254 nm, PMS, and UV/PMS systems, which follow the pseudo-first-order kinetics.

The aim of the present work is to analyze the possibility of decolorization of CV by different AOPs, such as UV-254 nm, PMS, and UV/PMS. Commonly referred to as AOPs, advanced oxidation processes are used to oxidize the complex organic in wastewater and that are difficult to decompose into simpler form end-products with biological processes [46,47].

## 2. Materials and Methods

### 2.1. Materials

The target pollutant under study is crystal violet (2-chloro-4-ethylamino-6-isopropylamino-s-triazine) (99.9%), which was purchased from Sigma Aldrich. The other chemicals, including potassium Peroxymonosulfate (PMS), sodium nitrate ($NaNO_3$), sodium nitrite ($NaNO_2$), sodium sulfate ($Na_2SO_4$), and sodium acetate ($CH_3COONa$), were purchased from Fisher Scientific. Sodium hydrogen carbonates ($NaHCO_3$), and sodium carbonate ($Na_2CO_3$) were purchased from Scharlau. To study the effect of pH, hydrochloric acid and sodium hydroxide, buffer solutions of 4.0 and 7.0 of sodium acetate were used to study the calibration process—these substances were purchased from Sigma Aldrich.

### 2.2. Analytical Methods

SpectroVis® Plus (OrderCodeGDX-SVISPL) was used for the evaluation of crystal violet under the degeneration of its mechanistic approach of photolysis, UV only, PMS only and homogenous UV/PMS processes. Go Direct SpectroVis Plus is a portable, visible-to-near-IR spectrophotometer and fluorometer; the instrument was equipped with the following specifications. The light source was incandescent with LED support, the detector was linear CCD, the wavelength range was usually 380–950 nm, the wavelength reporting interval was approximately ~1 nm, the optical resolution (FWHM) was in the range of 5.0 nm with a wavelength accuracy ± 4.0 nm, the photometric accuracy was ±0.10 A.U., the typical scan time was ~2 s, and the operating temperature was 15–35 °C [48]. The analysis was performed under ambient conditions by calibrating the Go Direct SpectroVis Plus with distilled water and a full-spectrum of the sample was taken in a cuvette about 3/4 full with the sample solution to be tested and then placed in the spectrophotometer where the spectrum absorbance vs. wavelength was obtained.

### 2.3. Experimental Set-Up for Photochemical Procedure

To carry out the oxidation process by means of an ultraviolet lamp, a schematic procedure was established. The experimental setup of this process usually involves a photochemical reactor along with a Petri dish made of Pyrex with dimensions of 60 mm (diameter) × 15 mm (height) with a quartz cover. Sample solutions were prepared with Milli-Q water (resistivity of 18.2 MΩ cm). The solutions of acidic and basic pH were prepared using the hydrochloric acid and NaOH solutions, respectively. In the end, solutions were examined for their TOC, where the sample solutions were allowed to irradiate at different intervals without the addition of a quenching compound. All experiments were carried out in triplicate.

## 3. Results and Discussion

### 3.1. Removal of CV by UV, PMS and UV/PMS Process

A comparative study for the removal of CV was carried out by UV-254 nm, PMS and UV/PMS system; the results are shown in Figure 1a. To investigate the effect of the oxidant, the degradation of CV was carried out in the presence of UV radiation only. Due to the presence of the pi-bonds in CV molecules, their degradation by UV radiation could be expected. However, only a 42% removal was observed under UV irradiation, reflecting its persistent nature. Similarly, only a 76% removal with PMS shows a higher removal efficiency compared to UV radiation only.

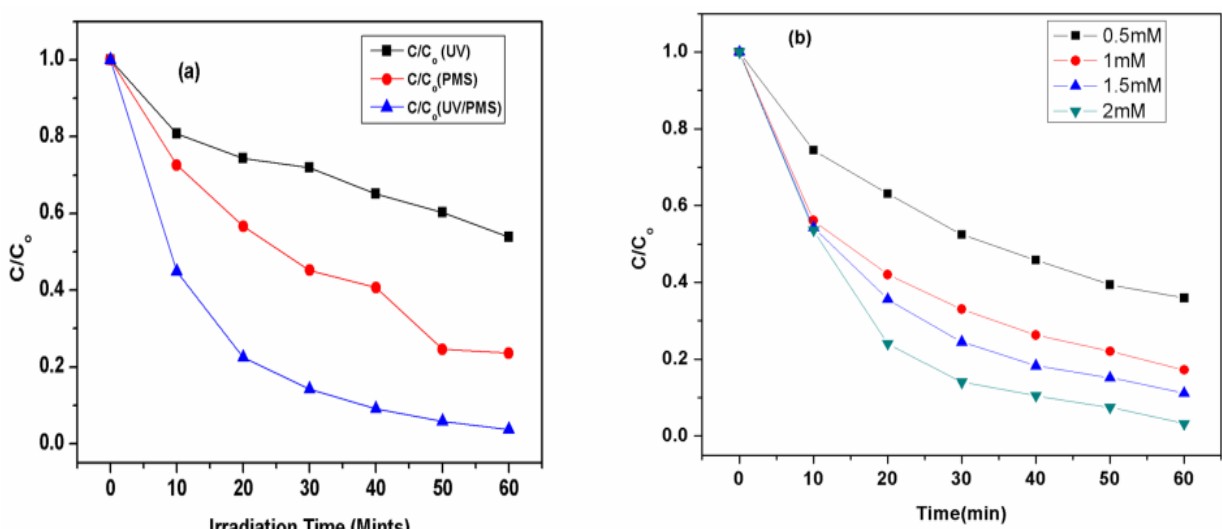

**Figure 1.** (**a**). Removal of CV by UV, PMS and UV/HSO$_5^{\bullet-}$ AOTs. Experimental conditions: [CV]$_\circ$ = 0.01 mM, [oxidant]$_\circ$ = 2.0 mM, [pH]$_\circ$ = 5.88 for PMS. (**b**). Effect of initial concentrations of PMS on its $K_{obs}$ $(mint)^{-1}$ and removal rate (mM $mint)^{-1}$ by UV/PMS process. Experimental conditions: [PMS]$_\circ$ = 2.0 mM, [pH]$_\circ$ = 5.88.

The coupling of PMS with a UV system significantly enhanced the degradation as well as the removal of CV up to 97%, which indicates the formation of reactive species such as $^{\bullet}$OH and SO$_4^{\bullet-}$, as shown in Equation (1) [49].

$$\text{HSO}_5^{\bullet-} + \text{hv} \rightarrow {}^{\bullet}\text{OH} + \text{SO}_4^{\bullet-} \quad \Phi \text{ of } {}^{\bullet}\text{OH} + \text{SO}_4^{\bullet-} = 1.04 \tag{1}$$

Figure 1a shows that higher removal efficiency of CV was obtained by UV/PMS, followed by the PMS and UV processes alone, respectively. The highest efficiency of UV/PMS process might be due to a higher quantum yield, such as ($\Phi$) of 1.04, as compared to the PMS system. Thus, there are two pathways for the generation of oxidant-free radicals

by the UV activation of the PMS. In the first pathway, which is considered the main pathway, there is a fission of an O-O bond by UV energy, as shown in Equation (2):

$$HSO_5{}^{\bullet-} + h\nu \rightarrow {}^{\bullet}OH + SO_4{}^{\bullet-} \tag{2}$$

The second pathway is where PMS is activated by electron conduction via the generation of electrons that interact with the UV radiation in water. The possible mechanism is shown in Equations (3) and (4):

$$H_2O + h\nu \rightarrow OH^{\bullet} + H^{\bullet} \tag{3}$$

$$HSO_5{}^{\bullet-} + H^{\bullet} \rightarrow SO_4{}^{\bullet-} + H_2O \tag{4}$$

In both mechanisms, the oxidant-free species generated both sulfate and hydroxyl radical attacks onto the pollutants and degraded them totally or partially, as shown in Equation (5) [50,51]:

$$SO_4{}^{\bullet-} + OH^{\bullet} + \text{organic compound} \rightarrow \text{organic by-products} + CO_2 + H_2O + SO_4{}^{\bullet 2-} \tag{5}$$

Various studies [52] showed that only those organic molecules can go through the process of direct photolysis when they fulfill two conditions: (1) all organic molecules should present in the ground state must absorb the radiation energy of suitable wavelength to undergo into an excited state, and (2) molecules in the excited state undergo a process of chemical disintegration, which is highly competitive with the non-excited state of molecules by physical infectiveness. This process can be represented by Equations (6)–(8) [53];

$$\text{Crystal Violet} \xrightarrow{h\nu} \text{Crystal Violet}^* \tag{6}$$

$$\text{Crystal Violet}^* \xrightarrow{h\nu} \text{Crystal Violet} \tag{7}$$

$$\text{Crystal Violet} \xrightarrow{h\nu} \text{Products} \tag{8}$$

This series of reactions shows that crystal violet undergoes degradation resulting in the formation of degradation of by-products along with $OH^{\bullet}$ radical formation, which further degrades the products of CV [52]. Thus, the removal of CV by reactive radicals depends upon the concentration of radicals, the nature of degradation products, and second-order rate constants.

### 3.2. Electrical Energy Comparison of the UV/PS, UV/PMS and UV/H$_2$O$_2$ Processes

The oxidation process is used for the removal of CV, and these processes are UV and UV/PMS and the peroxymonosulfate process. Their efficiency was determined by means of a comparison of their consumption of electrical energy per order (EE/O), which is defined as "the electrical energy in kilowatt hours (kWh) consumed for the removal of a particular pollutant by one order of magnitude in a unit volume (1 m$^3$) of polluted water". The results of the kinetics collected in a recent study may be applied to making the comparison of the EE/O of the studied AOPs, which may be deliberate by using the following Equation (9) [54].

$$EE/O = Pt/V \tag{9}$$

where, t is photolysis time (h), V represents total treated volume (m$^3$), and P is the total electrical power of the UV lamp (kW), correspondingly. The study of the removal of crystal violet by calculating their time of photolysis (t), which is needed to eliminate crystal violet by calculating the value of first- and fourth-order removals of crystal violet, usually given as $\ln(10)/(K_{obs})$ and $4 \times \ln(10)/(K_{obs})$, correspondingly. Their efficiency was measured in terms of their photolysis time, which is mandatory for the first-order removal of crystal violet.

The oxidation process brings about $1.0 \times 10^{-6}$ mM (0.000408 mg L$^{-1}$) removal of CV, which is a fourth-order removal of CV with a starting dose of 0.01 Mm (4.08 mg/L). The value of fourth-order removal indicates that this value is too small compared to the lethal dose of the CV present in freshwater used for drinking purposes [55]. Table 1 shows the data information about the first-order and fourth-order removal of CV through different processes, such as UV, PMS and UV/PMS systems for the oxidation of CV. These results indicate that UV/PMS is the most efficient and cost-effective process for the degradation of CV among the studied AOPs, as it has the lowest value of EE/O, and is a better system based on the results. The electrical energy for first-order removal and electrical energy for fourth-order removal value to UV/PMS system is lower, such as 1.68 and 6.48, respectively than PMS and UV only. Thus, the UV/PMS system is most economically suitable compared to the use of UV and PMS alone, as shown in Table 1.

**Table 1.** Electrical energy comparison of UV and UV/PMS processes for removal of CV. Experimental conditions: [CV]$_\circ$ = 0.01 mM and [PMS]$_\circ$ = 2 mM, [pH]$_\circ$ = 5.88.

| System | UV/PMS | PMS Only | UV Only |
|---|---|---|---|
| $K_{obs}$ (mints)$^{-1}$ | 0.2857 | 0.1269 | 0.0094 |
| Time (h)/O | 0.14 | 0.302 | 4.08 |
| EE/O (kWh m$^{-3}$/order) | 1.68 | 3.62 | 48.96 |
| Time (h)/four order | 0.54 | 1.210 | 16.33 |
| EE/fourth-order (kWh/m$^3$/order) | 6.48 | 14.52 | 195.96 |

*3.3. Effect of Various Initial Doses of Oxidant*

Table 2 and Figure 1b depict the effect of the initial dose of PMS to analyze the effect of peroxymonosulfate (HSO$_5$$^{\bullet-}$) on the degradation of *CV*, by altering the initial concentration of HSO$_5$$^{\bullet-}$ from 0.5 mM to 2 mM [56]. In order to study the kinetics for the degradation of crystal violet through UV/PMS, the observed pseudo-first-order rate constant ($K_{obs}$) was calculated from Equation (10) to (13), and the resulting values are depicted in Table 2.

$$ln\,(C/C) = K_{obs} \times t \tag{10}$$

$$-d\,[CV/dt] = k\,[CV] \tag{11}$$

$$-d\,[CV]/[CV] = k \times dt \tag{12}$$

$$-ln\,[CV]/[CV] = K_{obs} \times t \tag{13}$$

**Table 2.** The effect of the initial concentrations of PMS on the removal efficiency, $K_{obs}$, and degradation rate of CV dye by the UV/PMS system at [CV]$_\circ$ = 5 ppm, [PMS] = 0.5–2 mM (12 min).

| Reactor | Molar Ratio of [Oxidant]$_\circ$/[CV]$_\circ$ = 0.01 Mm (5 ppm) | Concentration (PMS) (mM) | %Removal = (C$_\circ$ − C) × 100/(C$_\circ$) | $K_{obs}$ (mints)$^{-1}$ | R$^2$ | Degradation Rate (mM/mints) | Time/Order Removal (min/O) | Time/Fourth-Order Removal (min/Fourth-Order) |
|---|---|---|---|---|---|---|---|---|
| | 50 | 0.5 | 64% | 0.094 | 0.953 | 0.057 | 24.522 | 98.087 |
| PMS | 100 | 1 | 83% | 0.160 | 0.918 | 0.071 | 14.400 | 57.601 |
| | 150 | 1.5 | 89% | 0.199 | 0.941 | 0.086 | 11.560 | 46.237 |
| | 200 | 2 | 97% | 0.286 | 0.974 | 0.099 | 8.059 | 32.238 |

These results showed that the degradation efficiency of CV increases with the increase in oxidant concentration, and the $K_{obs}$ obtained by using Equation (10) were also found to

increase with the increase in the initial concentration of PMS from 0.0939 to 0.2857 $\text{min}^{-1}$ for 0.5 to 2 mM, respectively, as shown in Table 2. These results were also consistent with findings in previously reported studies [56–58].

The possible increase of $k_{obs}$ ($\text{min}^{-1}$) is directly related to the generation of hydroxyl and sulfate radicals ($^{\bullet}\text{OH}$, $\text{SO}_4^{\bullet-}$), which increases with an increase in the rate of generation of both radicals when a higher $[\text{PMS}]_{\circ}$ was used, thus causing the increase in value of $k_{obs}$ at a high dose of PMS. It was also observed that there was a decrease of $k_{obs}$ when the initial dose of peroxymonosulfate was decreased due to the low concentrations of $^{\bullet}\text{OH}$ and $\text{SO}_4^{\bullet}$. To compute the time needed for first- and fourth-order removal of crystal violet, the initial dose of PMS was varied, as there have been implications of an observed pseudo-first-order rate constant, which is calculated by Equations (14) and (15):

$$\text{Time required for first-order removal (min/O)} = \ln10/k_{obs} \qquad (14)$$

$$\text{Time required for fourth-order removal (min/fourth-order)} = 4\ln10/k_{obs} \qquad (15)$$

To calculate the degradation rate for crystal violet, there is a need to define the degradation rate, which can be defined as "at different initial doses of peroxymonosulfate, there is a variation of concentration with respect to time (ppm/min)". In addition, the degradation rate was calculated for the initial duration of treatment, which is usually for 0 to 2 min. It was observed that the degradation rate was obtained to be 0.057 mM/min for [PMS] = 0.5 mM and 0.099 mM/min for [PMS] = 2 mM, as shown in Table 2. It was observed that the degradation rate increases when the initial dose of peroxymonosulfate [PMS] was increased because the high dose of peroxymonosulfate provides a higher rate of active radical generation, i.e., $^{\bullet}\text{OH}$ and $\text{SO}_4^{\bullet-}$, which causes the rapid rate of collision with the target molecules (crystal violet), which enhance the rate of degradation of *CV*. The results showed that the degradation efficiency of the CV dye increased when the concentration of the oxidants increased, as shown in Table 2. When the oxidant is at a low concentration, the degradation efficiency of CV is also low. This is because, at a low dose of peroxymonosulfate, the generation of hydroxyl and sulfate radicals will be lower, which is the main reason for the low value of $K_{obs}$ for the degradation of CV [56–58]. Others studies support these results [56,59]. Thus, the UV/PMS process is the most successful process for the degradation of crystal violet and the efficiency of such a process is further expanded with the study of different initial concentrations of CV, solution pH, and different inorganic ions.

The removal efficiency shown herein was determined after a 12 min reaction time. The degradation rate was determined for the initial reaction time from 0 to 2 min

### 3.4. Effect of Initial Dye Concentration

To investigate the practical applications of UV/PMS process and to find out the optimum dose of UV radiation applied to activate the PMS in $\text{UV}/\text{HSO}_5^{\bullet-}$ system for the efficient removal of crystal violet, the effects of the different initial doses of crystal violet was studied. Hence, the degradation rates and $K_{obs}$ for crystal violet were investigated at different initial doses of crystal violet ranging from 1–5 ppm at constant irradiation for 60 min and an initial dose of peroxymonosulfate [PMS] of 1 mM (Figure 2a). The results are shown in Table 3, which indicates that the maximum removal efficiency of crystal violet was 77% for 1 ppm when the reaction was run for 60 min, while there was only 42% removal of crystal violet for 5 ppm of the crystal violet. Moreover, other results in the literature [52,60,61] are consistent with these results. Peroxymonosulfate undergoes oxidation by ultraviolet radiation and produces active radicals, i.e., $^{\bullet}\text{OH}$ and $\text{SO}_4^{\bullet-}$, which cause the degradation of crystal violet, resulting in various kinds of degradation by-products (DPs), which are assumed to have high efficiency with hydroxyl and sulfate radicals. When the initial dose of crystal violet increased, there was a rise in the amount of degradation by-products, which created a competitive environment for active radicals of hydroxyl and sulfate. Thus, there is a competency between parent pollutants and

degradation by-products for both radicals. This competency will be more likely at high doses of parent pollutants compared to small doses of CV.

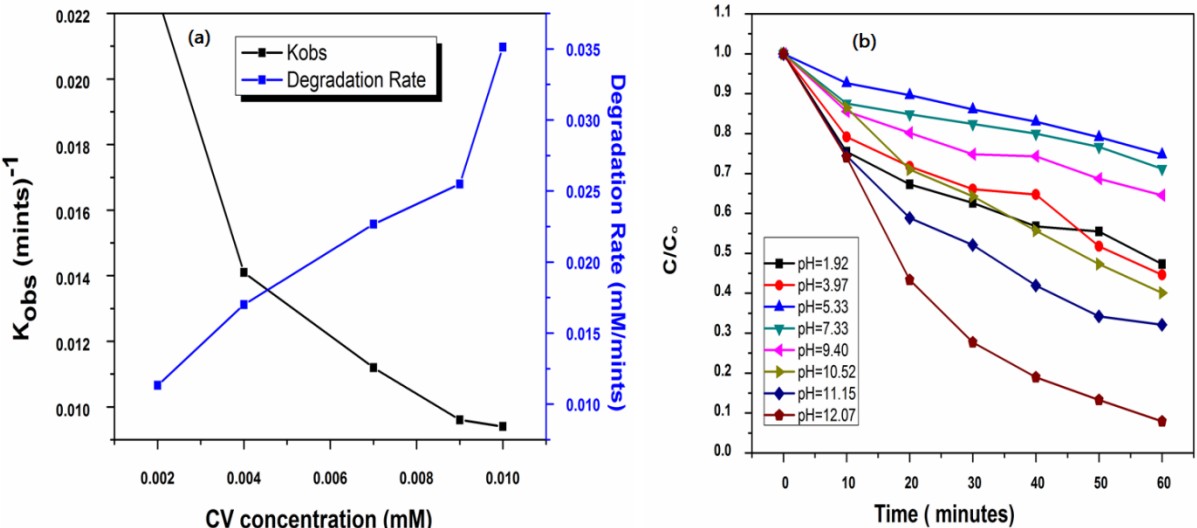

**Figure 2.** Effect of initial (**a**) concentration of CV on its $K_{obs}$ $(mint)^{-1}$ and removal rate (mM $mint)^{-1}$ by UV/PMS process at $[PMS]_{\circ} = 2.0$ mM, $[pH]_{\circ} = 5.88$ (**b**) solution pH on the removal efficiency of CV by U/PMS on removal efficiency of CV by UV/PMS process at $[CV]_{\circ} = 0.01$ mM (5 ppm), $[pH]_{\circ} = 1.92$–12.07.

The main cause of low degradation rates at higher concentrations of contaminants might be due to the molar ratios (proportion of peroxymonosulfate to CV), which reduce at lowers the concentrations of target contaminants and is considered the main cause of the lower degradation efficiency for CV at a higher dose compared to a small dose of CV [56,62]. For example, at 1 ppm, the molar ratio varies from 250 to 1000, but when the contaminant concentration is high, e.g., 5 ppm, the molar ratio range is 50–200.

The effect of the initial dose of crystal violet on the $k_{obs}$ was calculated, which was found to decrease with the rise in the initial dose of crystal violet. The results are shown in Table 3 and Figure 2a. These show that the value of $K_{obs}$ reduced from 0.0225 to 0.0094 $min^{-1}$ as the initial dose of crystal violet raised from 1 ppm to 5 ppm. These results are due to the high competency of crystal violet and their degradation by-products for active radicals along with the low ratio of crystal violet and of hydroxyl and sulfate radicals when the initial dose of crystal violet increased, which is the main cause for the low value of $k_{obs}$ with a high dose of crystal violet [44]. Another possible reason for lowering the removal efficiency of CV with an increase in $[CV]_{\circ}$ also increases the adsorption rate and consequently, light travels a smaller distance. Thus, a higher initial CV concentration reduces the proportion of irradiation energy required for the oxidation of PMS to generate the $SO_4^{\bullet-}$ formation by the filtering/shading effect. This results in a decrease in the value of $K_{obs}$ for CV through the ultraviolet-mediated peroxymonosulfate method. Similarly, the power function was obtained by a linear relationship between the initial dose of crystal violet and its observed pseudo-first-order rate constant. Moreover, the calculation of time that need to bring about the first-order and fourth-order removal of the target compound by using $k_{obs}$ at various doses of crystal violet by using Equations (15) and (16). The results are provided in Table 3.

Removal efficiency was determined at a 60 min reaction time, and the degradation rate for varying doses of crystal violet was determined for 0 to 2 min, which is 0.011 ppm/min for [CV] = 1 ppm and 0.035 ppm/min for [CV] = 5 ppm. The results are shown in Table 3. It is clear from these results that an increase in the initial concentration of CV resulted in an increase in the initial degradation rate because at a maximum dose of CV, a maximum amount of contaminants will be subjected to the $^{\bullet}OH$ and $SO_4^{\bullet-}$ active radicals, which might be a reason for the high degradation rate at a higher initial concentration of CV [56,57]. Other studies have reported similar results [43,56,58,59,63].

**Table 3.** Effects of the initial concentration of CV on the removal efficiency, $K_{obs}$ and degradation rate of CV dye by UV only. Experimental conditions: $[CV]_\circ$ = 1–5 ppm (0.002–0.01 mM), molar ratio when $[CV]_\circ$ = 1 ppm (0.002 mM).

| Reactor | Molar Ratio of $[Oxidant]_\circ/[CV]_\circ$ = 0.002 Mm (1 ppm) | Concentration (ppm) | Concentration (mM) | %Removal = $(C_\circ - C) \times 100/(C_\circ)$ | $K_{obs}$ $(mints)^{-1}$ | $R^2$ | Degradation Rate (mM/mints) | Time/Order Removal (min/O) | Time/Four-Order Removal (min/Fourorder) |
|---|---|---|---|---|---|---|---|---|---|
| | 250 | 1 | 0.002 | 77% | 0.0225 | 0.979 | 0.011 | 102.338 | 409.348 |
| | 500 | 2 | 0.004 | 62% | 0.0141 | 0.938 | 0.017 | 163.304 | 653.216 |
| Crystal violet | 750 | 3 | 0.007 | 50% | 0.0112 | 0.974 | 0.021 | 205.588 | 822.352 |
| | 1000 | 4 | 0.009 | 44% | 0.0096 | 0.977 | 0.025 | 239.853 | 959.410 |
| | | 5 | 0.01 | 42% | 0.0094 | 0.968 | 0.035 | 244.956 | 979.823 |

### 3.5. Effects of Initial pH

One of the most important factors in this process is pH, which controls the whole performance of advanced oxidation process. The pH values of wastewater vary widely from acidic to alkaline through to neutral. Thus, investigating the effect of pH in our study was considered vital for the potential applications of the UV-254/PMS system on a large scale-treatment of CV-contaminated waters due to the generation of active radicals of hydroxyl and $SO_4^{\bullet-}$ along with the chemical structure of CV. In addition to these effects, the variation in pH values directly affects the degradation of CV through UV/PMS. Thus, pH values play an important role in any chemical process involving the removal of pollutants from the water through an increase or decrease in its removal capability [64]. Hence, to examine the efficiency of the oxidation process, the effects of pH were studied in the range of 1.92–12.07 to explore the effect on the removal of CV by the UV/PMS system. The results are shown in Figures 2b and 3a.

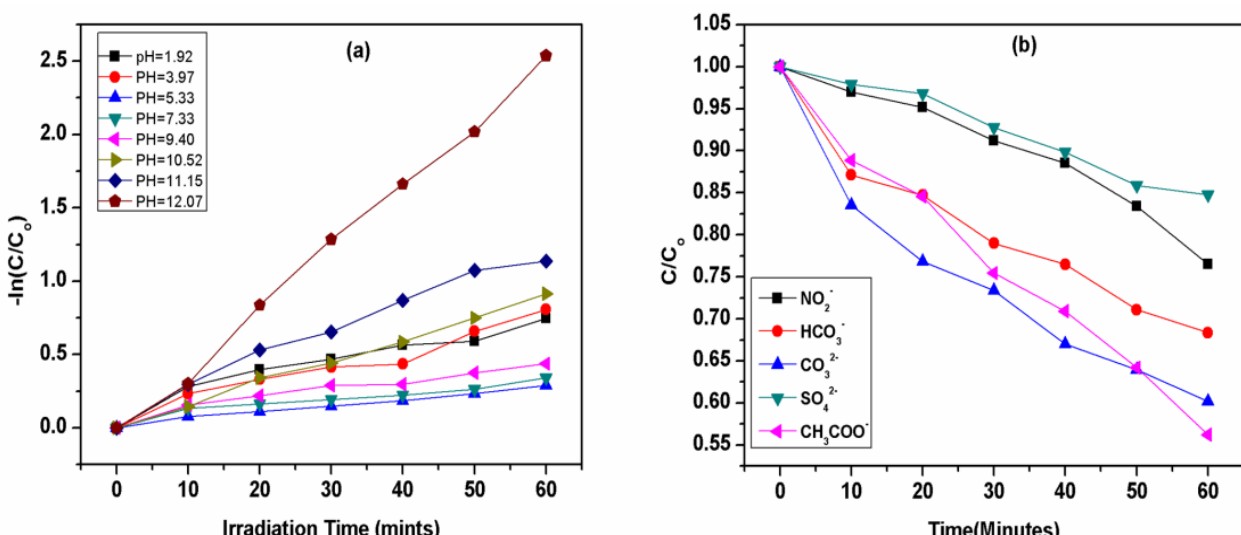

**Figure 3.** Effect of different (**a**) pH on degradation of crystal violet (**b**). inorganic anions ($NO_2^-$, $CO_3^{2-}$, $HCO_3^-$, $SO_4^{2-}$, $CH_3COO^-$) on the removal efficiency of CV by UV/PMS at $[CV]_o$ = 0.01 mM, $[NO_2]_o = [CO_3^{2-}]_o = [HCO_3^-]_o = [SO_4^{2-}] = [CH_3COO^-]_o$ = 1 ppm, $[pH]_o$ = 5.88.

The estimated degradation efficiency is determined by using Equation (16):

$$\text{Degradation Efficiency (\%)} = (C_o - C_t)/C_o \tag{16}$$

where $C_o$ = Initial concentration of dye at time t = 0, $C_t$ = Concentration of dye at time t.

It is clear from Figure 4 that an increase in pH initially decreases the removal efficiency and the maximum removal efficiency was obtained at pH 12.07. Crystal violet undergoes a decadence process at various pH levels by going through the series of initial pH = 12.07($K_{obs}$ = 0.0415 mints$^{-1}$) > pH = 11.15($K_{obs}$ = 0.0208 mints$^{-1}$) > pH = 10.52($K_{obs}$ = 0.0151 mints$^{-1}$) > pH = 1.92($K_{obs}$ = 0.0134 mints$^{-1}$) > pH = 3.97($K_{obs}$ = 0.0132 mints$^{-1}$) > pH = 9.40($K_{obs}$ = 0.0079 mints$^{-1}$) > pH = 7.33($K_{obs}$ = 0.0058 mints$^{-1}$) > pH = 5.33($K_{obs}$ = 0.0048 mints$^{-1}$). It is clear that the removal efficiency of CV first decreases from 53 to 29% at pH 1.92 to 7.33, then increases from 36 to 92% at pH 9.40 to 12.07, suggesting that at an acidic pH, the removal efficiency decreases while alkaline conditions favor the removal efficiency of CV dye. It is clear that pH plays an important role in the removal of pollutants by regulating the rate of generation of different radicals, such as the formation of the hydroxyl radical (OH$^{\bullet}$).

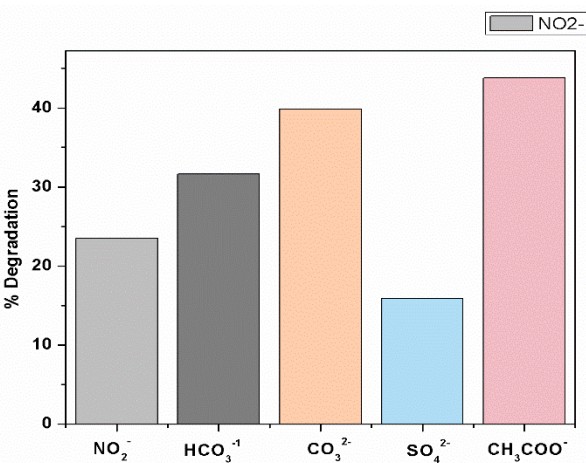

**Figure 4.** Effects of inorganic anions ($NO_2^-$, $CO_3^{2-}$, $HCO_3^-$, $SO_4^{2-}$, $CH_3 COO^-$) on the removal efficiency of CV by UV/PMS at $[CV]_\circ = 0.01$ mM, $[NO_2^-]_\circ = [CO_3^{2-}]_\circ = [HCO_3^-]_\circ = [SO_4^{2-}]_\circ = [CH_3COO^-]_\circ = 1$ ppm, $[pH]_\circ = 5.88$.

Figure 2b shows that there is the maximum removal of CV at extreme conditions of pH in both conditions, i.e., under highly acidic as well as under highly alkaline pH, which can be explained on the basis of the rate of formation of both types of ions and radicals under acidic and alkaline pH values, such as with $H^+$, $H^\bullet$, $OH^{\bullet-}$, and $^\bullet OH$ [65]. When the pH of the solution is acidic, such as 1.92–3.97, there is maximum availability of $H^+$ and $H^\bullet$, which shows the maximum degradation of CV, and hence, increases the removal efficiency of CV from an aqueous solution [66]. When the pH of the solution further increases from 5.33–7.33, the rate of degradation/decolorization of CV becomes low, which might be attributed to a decrease in the rate of formation of $H^+$ and $H^\bullet$ at this pH range, as shown in Figure 3a [67].

CV undergoes degradation under acidic pH by means of the following acidic mechanism of hydrolysis as shown in Equation (17):

$$CV + H^+ \rightarrow CVH^+ + H_2O \rightarrow CVOH + 2H^+ \tag{17}$$

It has been reported that there was a better removal efficiency of dyes with maximum degradation in solutions when the pH of the solution was maintained between 2.0–4.0. Similarly, in the UV/PMS system, the removal of CV is greatly influenced by the pH of the solution. The $H^+$ and $OH^{\bullet-}$ concentrations in the solution are greatly influenced by the change in the pH of the solution. Possible reasons for the decline in the removal efficiency of the dye include the scavenging of $^\bullet OH$ by $OH^{\bullet-}$, and the low redox potential of $^\bullet OH$. $^\bullet H$ and $^\bullet OH$ radicals reacts with $OH^{\bullet-}$, which are already at a low concentration due to their highly basic pH (i.e., 10.52, 11.15 and 12.02), where $OH^{\bullet-}$ are already in high concentration and thus, the effective concentration of $^\bullet H$ and $^\bullet OH$ [Equations (18) and (19)] radicals to reacting with the CV correspondingly decreases [68].

$$^\bullet H + HO^{\bullet-} \rightarrow e_{aq}- + H_2O \tag{18}$$

$$^\bullet OH + {}^{-\bullet}OH \rightarrow H_2O + O^{\bullet-} \tag{19}$$

The high removal efficiency of CV at pH 1.92 and 3.97 could also be due to lower scavenging of $e_{aq}-$ and $^\bullet OH$. Thus, in acidic conditions (pH 1.92 and pH 3.97), the recombination of $e_{aq}-$ and the $^\bullet OH$ radical is avoided (Equation (20)), thus allowing the $^\bullet OH$ radical to react with CV [69]:

$$e_{aq}^- + {}^\bullet OH \rightarrow OH^{\bullet-} \tag{20}$$

Moreover, the better removal efficiency at pH 1.92 and 3.97 might be due to the degradation of the crystal violet, which increases the elimination of crystal violet due

to the high exposure of $H_2O$ to UV radiation, which undergoes a disintegration process with the generation of $OH^{\bullet-}$, $H^+$, and $^{\bullet}OH$ radicals at high rates [70]. Secondly, the high degradation rate of CV with better removal capabilities at pH 1.92 and 3.97 can be attributed to the bubbling phenomenon of CV pollutants for hydroxyl radicals [58].

This can be attributed due to acidic conditions, i.e., (pH 5.33), wherein there is a significant possibility of a scavenging reaction of hydrogen ion ($H^+$) by hydroxyl radicals ($^{\bullet}OH$), which can be represented by Equation (21) [61,71–74]:

$$^{\bullet}OH + H^+ + e^- \rightarrow H_2O_2 \tag{21}$$

It is reported that at pH 5.33, there is a decline in the concentration of hydroxyl radicals ($^{\bullet}OH$) because of the generation of hydrogen peroxide, as shown in Equation (21), easily undergoes its disintegration process. Because of the disintegration of hydrogen peroxide, there will be a remarkable decrease in the formation of hydroxyl radicals due to formation of the complex compound with dye, which results in the lowering in removal efficiency when the pH is 5.33.

It has also been reported that when the pH of the solution is maintained in the range of 5.33 and 7.33, there is the formation of chloride ions from HCl, which have antagonistic conditions for the hydroxyl radicals, as shown in Equation (22). There will be a formation of $ClOH^{\bullet-}$ species with the reaction of chloride ions with hydroxyl radicals, which has a high tendency to react with $H^+$, resulting in the formation of $Cl^{\bullet}$. These $Cl^{\bullet}$ react with each other at a high rate with the formation of $Cl_2^{\bullet-}$, as shown in Equations (23)–(25). These $Cl_2^{\bullet-}$ have a very low tendency to react with organic compounds to make their degradation possible compared to $^{\bullet}OH$, which has a greater tendency towards organic compounds [51,52,75].

$$Cl^- + {}^{\bullet}OH \rightarrow ClOH^{\bullet-} \tag{22}$$

$$ClOH^{\bullet-} \rightarrow {}^{\bullet}OH + Cl^- \tag{23}$$

$$ClOH^{\bullet-} + H^+ \rightarrow Cl^{\bullet} + H_2O \tag{24}$$

$$Cl^- + Cl^{\bullet} \rightarrow Cl_2^{\bullet-} \tag{25}$$

In addition, at moderate basic pH, the formation of $HCO_3^-/CO_3^{2-}$ should be considered, as two well-known radical scavengers can reduce the degradation of the pollutant in the pH range of 7.33 to 9.33. Thus, due to the formation of these radical scavengers in the pH range of 7.33 and 9.33, the removal efficiency of CV dye due to the unavailability of $^{\bullet}OH$ and $^{\bullet}H$ is lowered.

$$CVH^+ + H_2O \rightarrow CVOH + 2OH^- + 2H^+ \rightarrow CVOH + 2H_2O \tag{26}$$

In the above Equation (26), CV stands for crystal violet, while $CVH^+$ is crystal violet in its protonic form.

It has been observed that the removal efficiency of dyes, along with their degradation process, is at a maximum when the pH of the solution is kept in the range of 10.50 to 12.07 [76]. The high removal of CV dye at an alkaline pH could be associated with the chemical instability of CV under an alkaline pH [77,78]. The possible reason for the high degradation of CV dye under these conditions is due to the remarkably high rate of the formation of hydroxyl radicals, which have a strong potential to react with the target pollutant, thus enhancing degradation efficiency [79–82].

Thus, the process of decomposition of CV under basic conditions can be represented by Equation (27)

$$CV + OH^- \rightarrow CVOH^- + H_2O \rightarrow CVH^+ + 2OH^- \tag{27}$$

### 3.6. Effect of Typical Inorganic Ions

The effects of typical inorganic ions were studied to examine their effect on removal efficiency. The inorganic anions commonly exist in natural water, which includes $Cl^-$, $NO_2^-$, $CO_3^{2-}$, $HCO_3^-$, $SO_4^{2-}$ and $CH_3COO^-$, etc., and make up the water system, which is generally found in the range of $10^{-5}$ to $10^{-3}$ M. It was observed that there is a high tendency of inorganic ions to react with hydroxyl and sulfate active radicals, as shown in Equations (28)–(37). Due to such features, the effect of ions such as nitrates, carbonates, bicarbonates, sulfate, acetates, etc., have been reported in terms of the degradation of crystal violet to demonstrate the efficiency of AOPs for the treatment of wastewater as well as for organic contaminants from industrial effluent.

Equations (30) and (34) show that the hydroxyl and sulfate radicals have a high tendency to react with sulfate and nitrate ions, and are usually considered to be the cause for the prohibition of efficient removal of crystal violet dye by the formation of inorganic ions. The decrease in the degradation rate was found in the following decreasing order: $SO_4^{2-} > NO_2^- > HCO_3^- > CO_3^{2-} > CH_3COO^-$. The degradation rate of CV is highly affected by radical scavenging counteracting the capability of hydroxyl radicals ($^\bullet OH$) and $e_{aq}^-$, which play major roles in the removal of dyes by increasing their removal efficiency. Thus, $CO_3^{2-}$ [61,83] and $NO_2^-$ ions are considered the most notable scavengers for hydroxyl radicals ($^\bullet OH$). $e_{aq}^-$, as represented in Equations (28), (30) and (31), respectively, subsequently influence the degradation rate by lowering the rate of reaction of hydroxyl radicals with CV [84,85].

The relative higher inhibition of CV degradation by $SO_4^{2-}$ and $NO_2^-$ ions could be due to its fast reaction with $^\bullet OH$ radicals, as shown by Equations (30) and (34). In the presence of these ions, there is competition between these ions and CV for an $^\bullet OH$ radical. Since the reaction between the $NO_2^-$ and $^\bullet OH$ radicals is very fast (Equation (30)), a smaller amount of the $^\bullet OH$ radical is available to react with CV [86], resulting in lower rates of degradation compared to the degradation in the absence of any additives (Figure 3b). Moreover, the greater inhibitory effect of $SO_4^{2-}$, and $NO_2^-$ is due to its high second-order rate constant with $^\bullet OH$ (reaction (30) and (34), which may have led to a greater competition of the $SO_4^{2-}$ and $NO_2^-$ with CV for $^\bullet OH$ and, consequently, higher inhibition was observed in the presence of $SO_4^{2-}$, and $NO_2^-$ in the present study.

In addition, $NO_2^-$ ions are also strong scavengers of $e_{aq}^-$, as shown in Equation (31) [58]. The removal efficiency of CV by UV/PMS was also inhibited to a greater extent in the presence of $HCO_3^-$ and $CO_3^{2-}$ ions. However, $HCO_3^-$ has a greater inhibiting effect than $CO_3^{2-}$. The removal of CV by $^\bullet OH$ was inhibited by $CO_3^{2-}$ but at a lower rate might be due to its low reactivity with $^\bullet OH$ (reaction 28). As the $HCO_3^-$ ions have faster kinetic energy towards $^\bullet OH$ (reaction (29), they are expected to decrease the probability of the reaction of $^\bullet OH$ with CV [27]. The addition of $CO_3^{2-}$ has been reported to have lower inhibiting effects than $SO_4^{2-}$, $NO_2^-$, and $HCO_3^-$ and, thus, $CO_3^{2-}$ is more efficient in the removal of CV dye than all other ions in the solution. The main cause of the efficient elimination of CV by $CO_3^{2-}$ is due to its low second-order rate constant with $^\bullet OH$ (reaction (32)), which might have led to the lower competition of the $CO_3^{2-}$ with CV for $^\bullet OH$ and a higher removal of CV by UV/PMS in the presence of $CO_3^{2-}$ and $HCO_3^-$. The effect of $CO_3^{2-}$ has been reported to raise the pH of a solution, and a higher pH has been reported to affect the reactivity of $^\bullet OH$ and enhances removal efficiency [87]. These results are supported by other studies reported in the literature [88,89].

A decrease in the removal efficiency in the presence of either of such anions is due to the scavenging effect of $^\bullet OH$ and $SO_4^{\bullet -}$. The results also show that $NO_2^-$ and $SO_4^{2-}$ have more effect on decreasing the removal efficiency and thus have more of an inhibition effect on the removal efficiency compared to $HCO_3^-$, $CO_3^{2-}$, and $CH_3COO^-$ [30,61]. The study of kinetics for these inorganic anions reported in the literature [30] and their corresponding

values are given in Equations (28) to (37) [30,90], and these results are also consistent with the present study [62].

$$\bullet OH + CO_3{}^{2-} \rightarrow CO_3{}^{\bullet-} + \bullet OH \rightarrow k = 3.9 \times 10^8 \ M^{-1} \ S^{-1} \tag{28}$$

$$\bullet OH + HCO_3{}^{-} \rightarrow CO_3{}^{\bullet-} + H_2O \rightarrow k = 8.6 \times 10^6 \ M^{-1} \ S^{-1} \tag{29}$$

$$\bullet OH + NO_2{}^{-} \rightarrow NO_2{}^{\bullet} + OH^{-} \rightarrow k = 8.0 \times 10^9 \ M^{-1} \ S^{-1} \tag{30}$$

$$NO_2{}^{-} + e_{aq}{}^{-} \rightarrow NO_2{}^{2-} \rightarrow k = 4.1 \times 10^9 \ M^{-1} \ S^{-} \tag{31}$$

$$SO_4{}^{\bullet-} + CO_3{}^{2-} \rightarrow CO_3{}^{\bullet-} + SO_4{}^{2-} \rightarrow k = 6.1 \times 10^6 \ M^{-1} \ S^{-1} \tag{32}$$

$$SO_4{}^{\bullet-} + HCO_3{}^{-} \rightarrow CO_3{}^{\bullet-} + HSO_4{}^{-} \rightarrow k = 2.8 \times 10^6 \ M^{-1} \ S^{-} \tag{33}$$

$$SO_4{}^{2-} + \bullet OH \rightarrow SO_4{}^{\bullet-} + OH^{-} \rightarrow k = 6.5 \times 10^7 \ M^{-1} \ S^{-} \tag{34}$$

$$\bullet OH + Cl^{-} \rightarrow ClOH^{\bullet-} \rightarrow k = 4.3 \times 10^9 \ M^{-1} \ S^{-1} \tag{35}$$

$$SO_4{}^{\bullet-} + Cl^{-} \rightarrow SO_4{}^{2-} + Cl^{\bullet} \rightarrow k = 3.0 \times 10^8 \ M^{-1} \ S^{-} \tag{36}$$

$$NO_2{}^{-} + e_{aq}{}^{-} \rightarrow NO_2{}^{2-} \rightarrow k = 4.1 \times 10^9 \ M^{-1} \ S^{-} \tag{37}$$

Figure 4 shows the effects of various ions, such as nitrates, bicarbonates, carbonates, sulfate and acetates in the concentration range of 1.0 mM for anions on the degradation of CV by a UV/PMS system. Among the other reported ions, only $SO_4{}^{2-}$ shows the suppression effect for the degradation of crystal violet. The inhibition effect of $SO_4{}^{2-}$ is mostly the extermination nature of hydroxyl and sulfate ions [91].

The removal efficiency and reaction time of the developed catalytic system are compared with other reported catalytic systems, and the results are shown in Table 4. These results showed that the presently developed catalyst system has a better removal efficiency in minimum reaction times and can serve as a promising catalyst for the removal of CV and other dyes.

**Table 4.** Comparison of removal efficiency for various dyes of different catalyst systems.

| Dyes | Oxidant/Treatments Methodology | Removal Efficiency (%) | Reaction Time (Min) | Reference |
|---|---|---|---|---|
| **Crystal violet** | UV/PMS | 97 | 12 | |
| | PMS | 76 | 12 | Present Study |
| | UV | 42 | 60 | |
| **Brilliant Green** | UV/PS | 63.1 | 30 | |
| | UV/PMS | 47.0 | 30 | [57] |
| | UV/$H_2O_2$ | 34.8 | 30 | |
| **Crystal Violet** | ZnO nanonails under UV irradiation | 95 | 70 | [92] |
| **Crystal Violet** | $MnO_2$-based nanofibrous mesh/photocatalyst | 97 | 90 | [67] |
| **Congo-red** | $H_2O_2$ coupled with nZVMn/PBC | 95 | - | [24] |
| | nZVMn/PBC | 77 | - | |

## 4. Conclusions

This study shows that UV/PMS AOTs are one of the most promising and auspicious advanced oxidation technologies for the decolorization of CV, which is directly affected by initial substrate concentration, initial oxidant concentration, pH and different inorganic ions. The comparative removal of CV has corresponding studies showing that better removal of CV has been carried out by a UV/PMS system rather than a PMS and UV system alone.

The better removal of CV by UV/PMS system is dependent on the formation of free radical species like $^{\bullet}OH$ and $SO_4^{\bullet-}$, which are formed by PMS by means of UV radiation of 254 nm. Only a 42% removal of CV dye was obtained by UV radiation, while only a 76% removal of CV dye was obtained by the PMS system. However, the coupling of PMS with the UV system significantly enhanced the degradation as well as the removal of CV dye by as much as 97%. The removal efficiency of crystal violet increases as the dosage of oxidant increases and vice versa.

It has been observed that the removal efficiency of CV decreases with the increase of the CV concentration. Meanwhile, the degradation rate increases when the initial concentration of CV increased then the $K_{obs}$ decreased. It is clear that $K_{obs}$ decreased first from 0.0134 to 0.0048 mints$^{-1}$ when the pH increased from 1.92 to 5.33, then the value of $K_{obs}$ becomes constant from 0.0048 to 0.0079 mints$^{-1}$ at pH 5.33 to 9.40, suggesting that the acidic condition was more auspicious to the decolorization of CV than neutral conditions. In addition, the value of $K_{obs}$ increases from 0.0079 to 0.0415 mints$^{-1}$ when the pH increases from 9.40 to 12.07, suggesting the best removal of CV dye at basic pH conditions and alkaline circumstances greatly accelerated the CV degradation. The maximum removal efficiency of crystal violet dye occurs at pH 12.07. The removal efficiency of CV is found to be 24%, 32%, 39%, 15%, and 44% for nitrates ions, $HCO_3^-$, carbonates ions, $SO_4^{2-}$, and $CH_3COO^-$, correspondingly. The best removal efficiency rate of CV dye was achieved with $CH_3COO^-$ and $CO_3^{2-}$ inorganic ions. The studies of the kinetics of the elimination of CV by UV/PMS at varying initial doses of PMS were demonstrated and concluded that the reaction succeeds through pseudo-first-order kinetics. For the elimination of toxic organic contaminants from water and for its conversion into less toxic substances, the method under study is more efficient and beneficial and less costly than other methods. Moreover, such systems can be developed for other pollutants/contaminants found in wastewater and can be installed directly on the site of flow from industrial sources. The present study also reinforces and suggests that future studies perform a comparison of different oxidation technologies and their efficiencies in improving the research scope.

**Author Contributions:** N.A. conducted the experiments and wrote and revised the manuscript, I.A.K. supervised and revised the manuscript, M.W. performed the visualization and validation of the manuscript, A.A.K. conducted the formal analysis in the manuscript, S.U.D. assisted with the methodology, S.A.Q. helped to review the manuscript, A.M.K. reviewed the manuscript and assisted with English editing, M.U.H. reviewed and edited the manuscript. All authors have read and agreed to the published version of the manuscript.

**Funding:** This research received no external funding.

**Data Availability Statement:** The data will be provided on request.

**Conflicts of Interest:** The authors have no conflict of interest for this study.

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
