# Peer review of "Activation of Peroxymonosulfate by UV-254 nm Radiation for the Degradation of Crystal Violet"

_water, doi:10.3390/w14213440_

Round 1

Reviewer 1 Report

The manuscript is critically reviewed and recommended for acceptance after addressing the following suggestions and comments.

1....Include all the major results in the abstract.

2....Revise highlights.

3.....What are the importance of peroxymonosulfate in swimming pools.

4....Can UV fluence rate affect degradation of Crystal violet. 

5....update the references and add the latest references of the similar studies, I am recommending some, e.g., Journal of Cleaner Production (2022) 134405 (http://dx.doi.org/10.1016/j.jclepro.2022.134405), Journal of Photochemistry and Photobiology A: Chemistry (2022) 114305(https://doi.org/10.1016/j.jphotochem.2022.114305), Journal of Water Process Engineering 49 (2022) 103130. 

Author Response

Reviewer 1

Comment

Response

Include all major results in abstract

Revised as suggested.

Revise highlights

Highlights have been revised and updated

What are the importance of peroxymonosulfate in swimming pools?

Peroxymonosulfate are non-chlorine based shock. They are used to oxidize any contaminants in the water while leaving chlorine or bromine sanitizers already present in the water.

  • Since no chlorine is added the swimming pool is available for swimming immediately after the shock has dissolved and time has been given for the oxidation process to complete.
  • Chlorine use can decrease, as less chlorine is needed to oxidize organic and inorganic matter in the pool.
  • Moreover PMS is ecofriendly in nature and considered as versatile in nature with high efficiency for the decolourization and decontamination of aquatic environment due to their excellent water solubility, low cost, ease of storage, easy for transportation, and low environmental impact, high rate of oxidation, simplicity, and no resultant sludge formation.

Can UV fluence rate affect degradation of crystal violet

Yes UV fluence rate affect the degradation of crystal violet but the different systems were compared at the similar treatment times to get evidence that which system have better efficiency at short treatment time.

Update the reference and add the latest reference of similar studies

Reviewer 1

Comment

Response

Include all major results in abstract

Revised as suggested.

Revise highlights

Highlights have been revised and updated

What are the importance of peroxymonosulfate in swimming pools?

Peroxymonosulfate are non-chlorine based shock. They are used to oxidize any contaminants in the water while leaving chlorine or bromine sanitizers already present in the water.

  • Since no chlorine is added the swimming pool is available for swimming immediately after the shock has dissolved and time has been given for the oxidation process to complete.
  • Chlorine use can decrease, as less chlorine is needed to oxidize organic and inorganic matter in the pool.
  • Moreover PMS is ecofriendly in nature and considered as versatile in nature with high efficiency for the decolourization and decontamination of aquatic environment due to their excellent water solubility, low cost, ease of storage, easy for transportation, and low environmental impact, high rate of oxidation, simplicity, and no resultant sludge formation.

Can UV fluence rate affect degradation of crystal violet

Yes UV fluence rate affect the degradation of crystal violet but the different systems were compared at the similar treatment times to get evidence that which system have better efficiency at short treatment time.

Update the reference and add the latest reference of similar studies

References have been updated and revised

Reviewer 2 Report

In this study “Activation of Peroxymonosulfate by UV-254 nm Radiation for 2 the Degradation of Crystal Violet”, authors designed the work systematically with performing some valuable experimental work. Major revision is recommended, and comments are listed below:

1.              English of this manuscript needs to be improved; many spelling errors, typing and grammatical mistakes exists in the whole manuscript.

2.              The introduction should be clarified in term of uniqueness and advantage what is the novelty of this work over the previous related work. The Introduction should be focused and goes over the topic. Why the authors think about using this method? What is new? What is the aim of this work; what are the gaps to cover. The authors need to cite relevant recent references including high impact journal to make the manuscript in broad range readers. Following recent references should be included in revision: Journal of Hazardous Materials, Volume 403, 2021, 123854; Environmental Research, Volume 207, 2022, 112609; Journal of Environmental Chemical Engineering, Volume 9, Issue 5, 2021, 106160.

3.              Why did the author select UV-254 nm.

4.              Why did authors select time of treatment instead of UV fluence.

5.              How can the UV fluence rate be determined.

6.              The author used high concentration of crystal violet, what is the reported concentration of the Crystal violet in the wastewater.

7.              Please suggest significance and prospect for future studies in the conclusion.

Author Response

English of this manuscript needs to be improved; many spelling errors, typing and grammatical mistakes exists in the whole manuscript.

Language of the paper has been improved and typographic mistakes have been removed

The introduction should be clarified in term of uniqueness and advantage what is the novelty of this work over the previous related work. The Introduction should be focused and goes over the topic. Why the authors think about using this method? What is new? What is the aim of this work; what are the gaps to cover. The authors need to cite relevant recent references including high impact journal to make the manuscript in broad range readers. Following recent references should be included in revision: Journal of Hazardous Materials, Volume 403, 2021, 123854; Environmental Research, Volume 207, 2022, 112609; Journal of Environmental Chemical Engineering, Volume 9, Issue 5, 2021, 106160.

Suggested changes incorporated and references updated in the bibliography

Why did the author select UV-254 nm?

The sample showed optimized results for UV-254nm and high reactive sulfate radicals can be generated by activation of PMS with UV-254nm radiation. Below this wavelength no degradation was observed.

Why did authors select time of treatment instead of UV fluence.

Three systems were compared at the same treatment time to get evidence that which system have better efficiency at short treatment time.

How can the UV fluence rate be determined.

NA

The author used high concentration of crystal violet, what is the reported concentration of the Crystal violet in the wastewater.

Our study reveals that all experiments were conducted at 5 ppm initial concentration of CV. Reported concentration of crystal violet is usually range from 100-500 ppm in waste water. (https://doi.org/10.1016/j.reffit.2017.01.009)

Please suggest significance and prospect for future studies in the conclusion

Modified as suggested

Author Response

Some spelling/grammatical errors need to be corrected throughout the paper.

All spelling/grammatical mistakes have been rectified

Abstracts suggest starting with the background and then narrowing down the relevant topics that will be studied in the paper. Maybe this part can be improved.

Abstract is modified and improved

The authors should compare this work with others to highlight the novelty.

A new table no. 4 has been added which shows the comparison of present results with reported data

As shown in Figure 2b, for the analysis of the effect of different pH on the degradation effect, the authors are advised to refer to and cite some related papers for explanation.

Proper references have now been added.

Chemical formula subscripts should be standardized throughout the manuscript, such as the writing of ions in Figure 3b. Please check the manuscript carefully.

Modified as suggested and chemical formula subscripts have been modified throughout the manuscript.

Reviewer 4 Report

After reading the manuscript I have following remarks.

1. General comment. The paper must be read and corrected by a native speaker. In many places the phrases are confusing and misleading.

2. Line 15, 19 etc.  What was the accuracy of the pH measurement? 0,01? It should be corrected. 

3. Line 104 and section 3.7. It is not clear if authors identyfied CV degradation products by HPLC-MS. If not, why authors mentioned about it? there are no data about HPLC-MS in section methods. It is highly misleading.

4. Authors informed "Figure 5. Proposed degradation pathways for Crystal Violet (CV) by use of HPLC-MS analysis in UV/PMS system" What does it mean in real? The article does not describe the method or results of HPLC-MS analysis. It is highly misleading.

5. Section 2.2. What were LOD and LOQ of CV by spectrohpotometric method applied in the study? It should be added. There are no information about HPLC-MS study.

6. Section 2.3. It is not known how authors conducted the experiments. What kind of lamp, what power, what initial concentration of crystal violet, was used etc. It must be added and clearly described. The picture of the experimental setup should be added.

7. Section 2.3 Why authors use perchloric acid for pH value correction??? Perchloric acid is one of the strongest oxidizing acids!!! It can oxidize crystal violet!!!

I have included just a few of my comments above. Due to the large amount of misleading information, carelessness and a lack of scientific accuracy, I recommended that the manuscript should be rejected.

Author Response

Reviewer 4

General comment. The paper must be read and corrected by a native speaker. In many places the phrases are confusing and misleading.

Language is improved and paper is edited

Line 15, 19 etc.  What was the accuracy of the pH measurement? 0,01? It should be corrected.

It is 0.01 and is now corrected.

Line 104 and section 3.7. It is not clear if authors identified CV degradation products by HPLC-MS. If not, why authors mentioned about it? there are no data about HPLC-MS in section methods. It is highly misleading.

HPLC-MS was discussed as a technical type cause and we have deleted the proposed degradation pathway and analysis by HPLC-MS in the revised manuscript.

Authors informed "Figure 5. Proposed degradation pathways for Crystal Violet (CV) by use of HPLC-MS analysis in UV/PMS system" What does it mean in real? The article does not describe the method or results of HPLC-MS analysis. It is highly misleading.

HPLC-MS was discussed as a a technical type cause and are not the real experimental analysis by HPLC-MS and we have deleted the proposed degradation pathway and analysis by HPLC-MS in the revised manuscript and is now corrected in the revised manuscript.

Section 2.2. What were LOD and LOQ of CV by spectrohpotometric method applied in the study? It should be added. There are no information about HPLC-MS study.

HPLC-MS is not used for analysis and so there is no need of such informations.

Section 2.3. It is not known how authors conducted the experiments. What kind of lamp, what power, what initial concentration of crystal violet, was used etc. It must be added and clearly described. The picture of the experimental setup should be added.

The required details of UV lamp and initial concentration was provided in section 2.2, 2.3 and etc.

Section 2.3 Why authors use perchloric acid for pH value correction??? Perchloric acid is one of the strongest oxidizing acids!!! It can oxidize crystal violet!

In section 2.3. Hydrochloric acid was used for pH correction and perchloric acid was reported by mistake and is now corrected.

Round 2

Reviewer 1 Report

The authors have addressed all my comments and thus recommending acceptance of the revised manuscript. 

Reviewer 2 Report

Authors have made required changes. Manuscript can be accepted now.

Reviewer 3 Report

It can be accepted.

Reviewer 4 Report

I have doubts as to whether the article has been corrected by a native speaker.

E.g. "ejection to untreated water" In my oppinion we can "enter wastewater to the environment etc."

"the study dedudes", we can deduce sth from the study or sth can be deduced from the study

lines 32-34, totally incorect and misleading sentence

lines 196-197 "molecules in the excited state undergoes into process of chemical disintegration which is highly competitive one with the non-excited state of molecules by physical infectiveness. (dot???) which can be represented by following process such as Eq. (6 - 8) [53];"

and many others.